# A systematic, complexity-reduction approach to dissect the kombucha tea microbiome

Xiaoning Huang[1,2,3], Yongping Xin[1,2], Ting Lu[1,2,4,5,6]*

[1]Department of Bioengineering, University of Illinois Urbana-Champaign, Urbana, United States; [2]Carl R. Woese Institute for Genomic Biology, University of Illinois Urbana-Champaign, Urbana, United States; [3]College of Food Science and Nutritional Engineering, China Agricultural University, Beijing, China; [4]Department of Physics, University of Illinois Urbana-Champaign, Urbana, United States; [5]Center for Biophysics and Quantitative Biology, University of Illinois Urbana-Champaign, Urbana, United States; [6]National Center for Supercomputing Applications, Urbana, United States

**Abstract** One defining goal of microbiome research is to uncover mechanistic causation that dictates the emergence of structural and functional traits of microbiomes. However, the extraordinary degree of ecosystem complexity has hampered the realization of the goal. Here, we developed a systematic, complexity-reducing strategy to mechanistically elucidate the compositional and metabolic characteristics of microbiome by using the kombucha tea microbiome as an example. The strategy centered around a two-species core that was abstracted from but recapitulated the native counterpart. The core was convergent in its composition, coordinated on temporal metabolic patterns, and capable for pellicle formation. Controlled fermentations uncovered the drivers of these characteristics, which were also demonstrated translatable to provide insights into the properties of communities with increased complexity and altered conditions. This work unravels the pattern and process underlying the kombucha tea microbiome, providing a potential conceptual framework for mechanistic investigation of microbiome behaviors.

*For correspondence:
luting@illinois.edu

## Editor's evaluation

This work will be of interest for researchers studying the functions of microbial communities, microbial ecology and interactions. Using the Kombucha tea (KT) microbiome as a case study, Huang et al., provide a framework for simplifying complex communities into core communities that capture aspects of complex communities. Authors demonstrated that core communities can facilitate a mechanistic understanding of how microbes interact, especially when member species are individually culturable. The work presents a fresh, novel approach for the coarse-grained analysis of complex microbiomes.

## Introduction

Microbiome populates the planet Earth, driving the growth of plants (*Bulgarelli et al., 2013*; *Singh et al., 2020*) biogeochemical cycling of elements (*Falkowski et al., 2008*; *Canfield et al., 2010*), and health and disease of humans (*Cho and Blaser, 2012*; *Turnbaugh et al., 2007*). Over the past decades, microbiome has gained explosive interest across disciplines from both academia and industry. To date, most efforts have focused on species cataloging (*Oliverio et al., 2018*), composition-phenotype

association (*Zhernakova et al., 2016*; *Trivedi et al., 2020*) and microbiome-environment correlation (*Bahram et al., 2018*). These efforts yielded invaluable insights into ecosystem structure and function, reinforcing the need for microbiome research. Moving forward, an overarching goal is to dissect microbiome causation and mechanism (*Fischbach, 2018*; *Gilbert et al., 2018*). Specifically, required to be uncovered are the causes of specific microbiome traits and underlying mechanisms that drive the emergence of these traits. Tackling this challenge is important, because it will help to understand community structure and dynamics, predict the impacts of microbiome on habitats and design interventions for modulating ecosystem function (*Schmidt et al., 2018*; *Skelly et al., 2019*).

To achieve the goal, one promising path is to dissect the metabolic underpinnings of members constituting a microbiome. Metabolism is a defining cellular process through which microbes acquire nutrient and energy; thus, its characteristics determine the growth of individual species. Through metabolism, cells also produce substances that are beneficial or deleterious to the growth of other species. Additionally, metabolism is often accompanied with the production of biomolecules that are bioactive to habitats (e.g. human, soil, and plant). These molecules directly affect habitats, for instance, short-chain fatty acids produced by the gut microbiome shape the immune function and brain behavior of human (*Kau et al., 2011*; *Dalile et al., 2019*). Alternatively, they may remodel the physiochemical properties of the habitats, through which microbiome realizes indirect functional modulation. For example, extracellular polysaccharides secreted by probiotic bacteria trigger biofilm formation in the gastrointestinal tract, which promotes the host' resistance to infection (*Lebeer et al., 2011*). Thereby, targeting microbial metabolic underpinnings offers a systematic route to decode microbiome composition and function.

The pursuit of this path is, however, hindered by the intrinsic, remarkable complexity of native ecologies. For instance, the human gut microbiome consists of over 1000 species and 100 trillion cells (*Qin et al., 2010*) a teaspoon of healthy soil contains over 10,000 taxa members totaling up to 1 billion cells (*Kumar et al., 2021*). To circumvent the challenge, researchers have recently turned to microbiome cores (*Shade and Handelsman, 2012*; *Risely, 2020*; *Hernandez-Agreda et al., 2017*), simplified communities that are abstracted from native ecosystems but retaining their key structural and functional characteristics. Supporting the notion, studies have revealed a core gut microbiome across human population regardless of body weight (*Turnbaugh et al., 2009*). Additionally, across soda lakes separated in distance, there is a collection of common microbes with similar structural patterns (*Zorz et al., 2019*). These simplified systems are approximations of native communities, providing a powerful alternative to study complex ecosystems.

Here, we hypothesize to interrogate metabolic underpinnings of minimal cores as a causal and mechanistic strategy to elucidate microbiome structure and function. To test the hypothesis, we adopted the kombucha tea (KT) microbiome as our model ecosystem. Commonly called a symbiotic culture of bacteria and yeasts (SCOBY) (*Soares et al., 2021*), the microbiome drives the fermentation of KT, a slightly sweet, acidic beverage with multiple health benefits (*Jayabalan et al., 2014*; *Villarreal-Soto et al., 2018*). During the fermentation, the microbiome also produces floating pellicles at the air-liquid interface (*Laavanya et al., 2021*). Compared to microbial ecologies in the soil and the human body, the KT microbiome is relatively simple in composition, easy to cultivate and amendable for quantification. Additionally, it involves species that are well characterized and feasible for perturbations. In fact, food microbiomes including those in kefir grain (*Blasche et al., 2021*), cheese rind (*Wolfe et al., 2014*; *Zhang et al., 2018*; *Cosetta and Wolfe, 2020*; *Pierce et al., 2021*), wine (*Melkonian et al., 2019*), and kimchi (*Miller et al., 2019*) have been lately exploited as tractable platforms for studying community diversity, succession, and niche partition (*Wolfe and Dutton, 2015*).

Our specific research started by characterizing the composition and metabolite patterns of the microbiome from commercially available KT drinks. We then used isolates to assemble 25 two-species consortia from which a minimal core was identified. Temporal fermentation showed that the core was convergent in its population composition, coordinated on temporal patterns of metabolites and capable of pellicle formation. Through comprehensive culturing of individual species under defined substrates, we obtained a casual and mechanistic understanding for the observed structural and functional traits of the core. We further showed that the knowledge from the core was translatable to account for the properties of communities with increased complexity and altered conditions. Together, our work illustrates the pattern and process underlying the composition and function of the

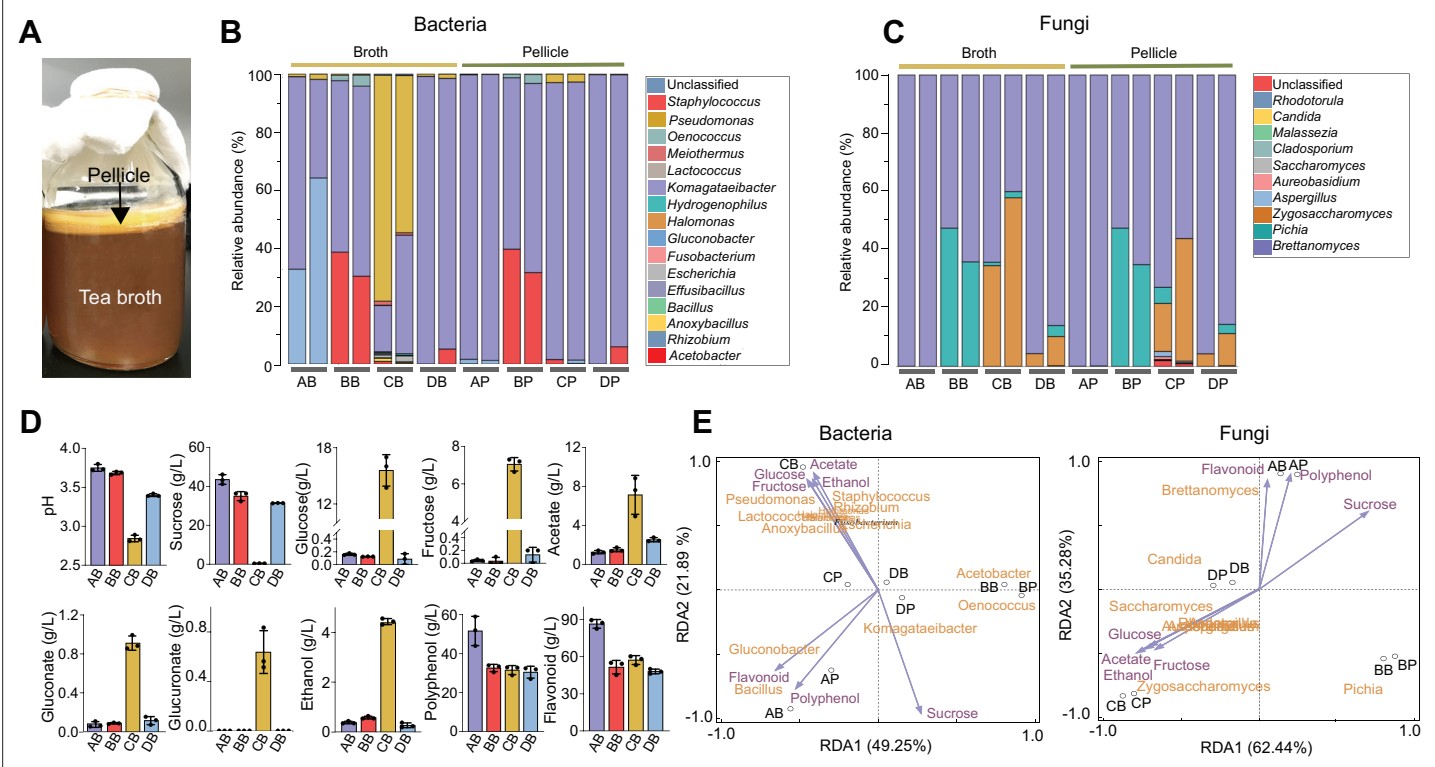

**Figure 1.** Characterization of the native KT microbiome. (**A**) Image of a typical kombucha tea fermentation containing both broth and pellicle. (**B, C**) Microbial composition in the broths and pellicles of different kombucha teas at the bacterial (**B**) and fungal (**C**) genus levels. For the four tea samples (A, B, C and D), their broths are named AB, BB, CB, and DB whereas their pellicles are called AP, BP, CP and DP respectively. For each sample, two duplicates are presented. (**D**) Chemical properties of the kombucha tea broths at the end of fermentation (day 14). Measured variables include pH, sucrose, glucose, fructose, acetate, gluconate, glucuronate, ethanol, polyphenol, and flavonoid. Bars and error bars correspond to means and s.d. (**E**) Correlation between microbial composition with biochemical substances in the KTs uncovered by redundancy analysis. Purple arrows represent metabolites, black circles represent different tea samples.

KT microbiome, providing a promising conceptual framework for mechanistic investigation of microbiome behaviors.

# Results

## Characterization of the native KT microbiome

We set out to identify key structural and functional traits of the KT microbiome by performing fermentations with commercially available SCOBYs and black tea substrate supplemented with 50 g/L of sucrose (Materials and methods). Each of the fermentations resulted in a light-brown broth and a floating, gel-like pellicle (*Figure 1A*), which were analyzed in terms of their compositional diversity and metabolite abundance using amplicon sequencing and high-performance liquid chromatography respectively. Here, we considered metabolite profiles as a representation of microbiome function because chemical ingredients in KT broth are key factors conferring benefits (*Soares et al., 2021*; *Villarreal-Soto et al., 2018*).

Our results showed that the microbiome had a relative low diversity, dominated by four bacterial genera, namely *Komagataeibacter*, *Acetobacter*, *Gluconobacter*, and *Pseudomonas* (*Figure 1B*), and three fungal genera including *Brettanomyces*, *Pichia*, and *Zygosaccharomyces* (*Figure 1C*). The bacteria and fungi also exhibited different context dependences: the composition of the former could vary significantly between the broth and pellicle of a single KT sample, such as samples A and C (AB (sample A's broth) vs. AP (sample A's pellicle), CB vs. CP) (*Figure 1B*); by contrast, the composition of the latter remained consistent across broth and pellicle (*Figure 1C*). Additionally, for bacteria, *Komagataeibacter* was the overall most predominant genus across samples and other genera were

prevalent only in selected cases. For example, *Acetobacter* was prevalent in BB and BP, *Gluconobacter* was dominant in AB and *Pseudomonas* was predominant in CB. For fungi, *Brettanomyces* was predominant in all samples but, in samples B and C, *Pichia* and *Zygosaccharomyces* were also widespread. Thus, bacteria and fungi both served as constituting members of the microbiome, with *Komagataeibacter* and *Brettanomyces* being the dominant bacterial and fungal genus accordingly. This compositional pattern was consistent with previous reports although *Pseudomonas* was typically low in abundance (*Reva et al., 2015*; *Villarreal-Soto et al., 2020*).

In parallel, we quantified the biochemical characteristics of KT broths, including pH, sugars, acids, and tea-derived substances (*Figure 1D*). The final pH values of samples A, B, D were around 3.6 while the pH of sample C was 2.8, all of which were in the reported range of a matured KT safe for human consumption (*Cardoso et al., 2020*). The sucrose concentration dropped from 50 g/L to 30–40 g/L except for sample C whose sucrose was depleted. There were also trace amounts of glucose and fructose except for sample C containing a high level of the sugars. Acetate, gluconate, and glucuronate were also detected, among which acetate had the highest concentration. Again, sample C was the outlier with a much higher level of acids. Since the concentration of gluconate was relatively low in our experiment and varied greatly across previous studies (*Jayabalan et al., 2014*; *Coton et al., 2017*), we would not consider it as a characteristic metabolite. The fermentation also resulted in the accumulation of ethanol (~0.5 g/L for samples A, B, and D and 4.4 g/L for sample C). Two tea-derived compounds, polyphenol and flavonoid, were abundant (~30 g/L and ~50 g/L respectively). To reveal how these metabolites correlate with microbial composition, we performed redundancy analysis over the four samples (*Figure 1E*). Notably, because bacterial and fungal species were calculated separately during the amplicon sequencing, we separated the analysis of microbe-metabolite correlation for bacteria and fungi accordingly. Nevertheless, the results showed that acetate, ethanol, glucose, and fructose are co-localized with each other but oppositely located compared to sucrose, polyphenol, and flavonoid are close to each other but remote from all other metabolites, and these metabolites have different KT microbiome samples close by, suggesting there are intrinsic correlations among metabolites and between metabolites and microbes.

From the above results, we drew three traits as the defining characteristics of the KT microbiome: first, it involves both bacteria and yeasts; second, it consumes sucrose with the synthesis of acetate, ethanol and a low level of glucose and fructose as the primary extracellular metabolites; third, it results in pellicle formation. These traits serve as the criteria for the identification of a proper microbiome core.

## Selection of a minimal core for the KT microbiome

To develop a correct core that recapitulates the native microbiome, we isolated a series of strains from the KT samples (*Figure 2—source data 1*). From the isolates, we selected five bacterial species, including *Komagataeibacter rhaeticus* ($B_1$), *Komagataeibacter intermedius* ($B_2$), *Gluconacetobacter europaeus* ($B_3$), *Gluconobacter oxydans* ($B_4$) and *Acetobacter senegalensis* ($B_5$), and 5 fungal species, including *Brettanomyces bruxellensis* ($Y_1$), *Zygosaccharomyces bailii* ($Y_2$), *Candida sake* ($Y_3$), *Lachancea fermentati* ($Y_4$) and *Schizosaccharomyces pombe* ($Y_5$), for synthesizing microbiome cores. Guided by the criterium that the KT microbiome contains both bacteria and fungi, we performed combinatorial mixing of the selected isolates, resulting in 25 two-species minimal core candidates with each involving one bacterial and one fungal species. To determine whether these candidates resemble the native, we conducted KT fermentation with these candidates and their corresponding 10 monocultures and, subsequently, quantified their microbial composition, extracellular metabolites and pellicle formation (Materials and methods).

From colony forming units (CFU) counting (*Figure 2A*, *Figure 2—source data 2*), we found the bacterial and fungal species coexisted in all co-cultures as in the native KT microbiome. Additionally, in most cases, bacteria and yeasts had comparable relative abundances (<10 folds of difference) except for the combinations $B_2Y_3$ and $B_4Y_3$ whereby the bacteria were 100 times less than the yeast, suggesting these two combinations might not be the best candidates. For monocultures, the bacteria showed highly variable CFU while the yeasts yielded comparable CFU, indicating that bacteria varied greatly in sucrose utilization while yeasts were all stably capable.

By measuring pH, sugars, acids and tea-derived substances in the broths, we also obtained the biochemical characteristics of the candidates (*Figure 2B*, *Figure 2—source data 3*). The results

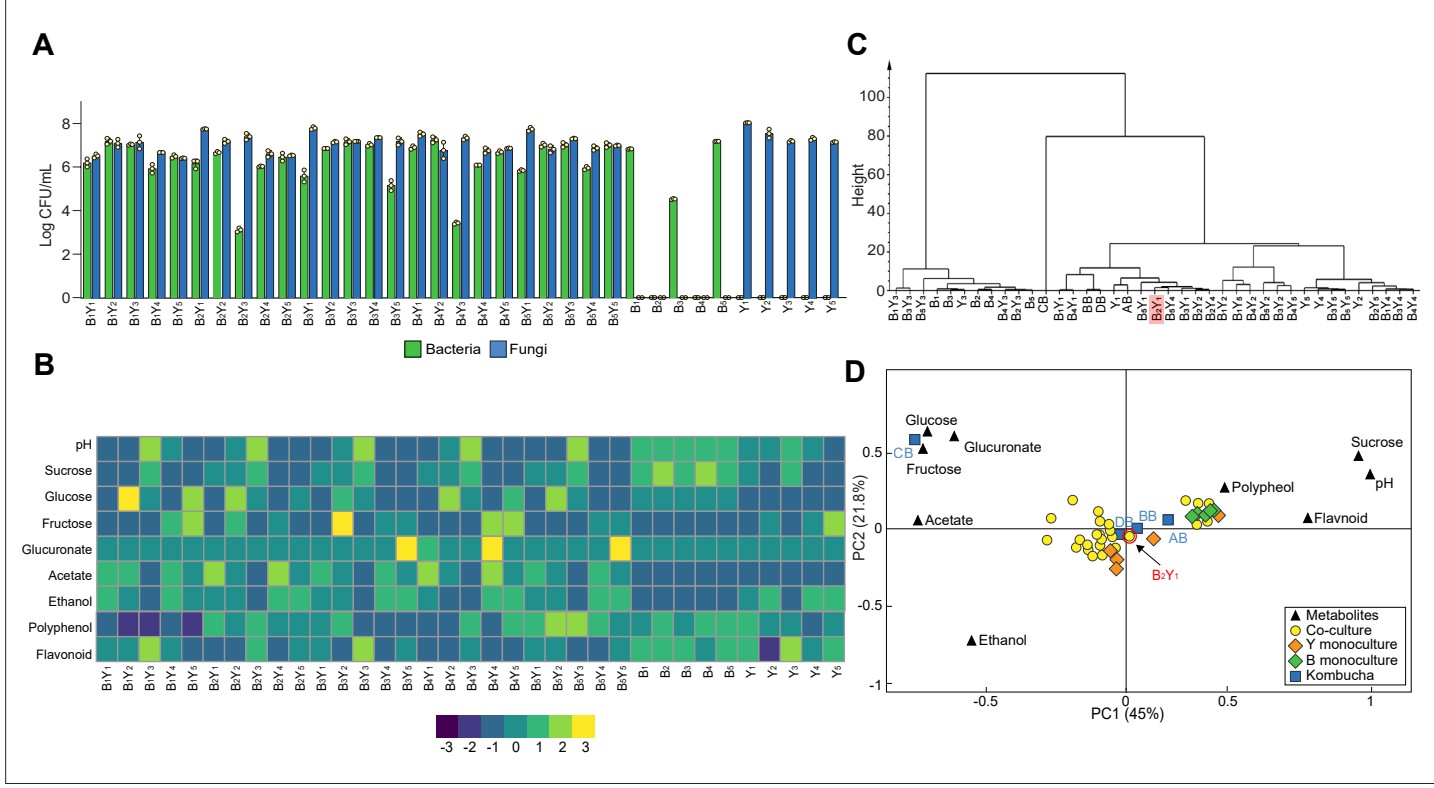

**Figure 2.** Population and metabolic quantification of two-species core candidates. (**A**) Colony forming units (CFU) counting of 25 two-species core candidates and 10 monoculture controls upon fermentation. Each core candidate is composed of one bacterial and one fungal species selected from the 10 isolates: $B_1$ (*Komagataeibacter rhaeticus*), $B_2$ (*Komagataeibacter intermedius*), $B_3$ (*Gluconacetobacter europaeus*), $B_4$ (*Gluconobacter oxydans*), $B_5$ (*Acetobacter senegalensis*), $Y_1$ (*Brettanomyces bruxellensis*), $Y_2$ (*Zygosaccharomyces bailii*), $Y_3$ (*Candida sake*), $Y_4$ (*Lachancea fermentati*), and $Y_5$ (*Schizosaccharomyces pombe*). Each monoculture control is one of the ten isolates. (**B**) Chemical property analysis of the core candidates and their controls. Measured variables include pH, sucrose, glucose, fructose, glucuronate, ethanol, acetate, polyphenol, and flavonoid. Here, the heatmap illustrates the relative concentrations of metabolites compared to their means across all samples using the Z-score normalization (***Quan et al., 2019***). (**C**) Hierarchical cluster analysis of the metabolic properties of the samples. The candidate $B_2Y_1$ is highlighted. (**D**) Principal component analysis of the metabolic properties. The candidate $B_2Y_1$ is circled in red.

The online version of this article includes the following source data for figure 2:

**Source data 1.** List of isolated bacterial and fungal species.

**Source data 2.** Counting data of different bacteria and yeasts co-cultures.

**Source data 3.** Chemical property analysis of the core candidates.

**Source data 4.** Statistics of HCA and PCA analysis.

showed that the co-cultures had comparable pH (~3.5) except for the five involving $Y_3$. The $Y_3$-involving candidates also yielded a significantly higher level of residual sucrose and a significantly lower level of acetate and ethanol compared to others, suggesting that these candidates were unsuitable to serve as cores. The metabolite profiles of the monocultures showed that the yeasts alone could be sufficient for sucrose consumption. It also showed that acetate was produced primarily through co-cultures but not monocultures. To systematically evaluate the candidates, we performed hierarchical cluster analysis and principal component analysis over the metabolites to determine the similarities among the candidates and the four native samples (AB, BB, CB, and DB). The hierarchical cluster analysis yielded three groups, one involving bacteria monocultures and $Y_3$-involved mono- and co-cultures, another containing CB only, and the third including the rest (***Figure 2C***, ***Figure 2—source data 4***). The principal component analysis showed that the co-cultures were all relatively close to the native microbiomes except for CB (***Figure 2D***).

We further evaluated the candidates in terms of pellicle formation, the third characteristic of the native microbiome. The results showed that five co-cultures, $B_2Y_1$, $B_2Y_2$, $B_2Y_3$, $B_2Y_4$, and $B_2Y_5$, successfully produced pellicles during sucrose fermentation (data not shown).

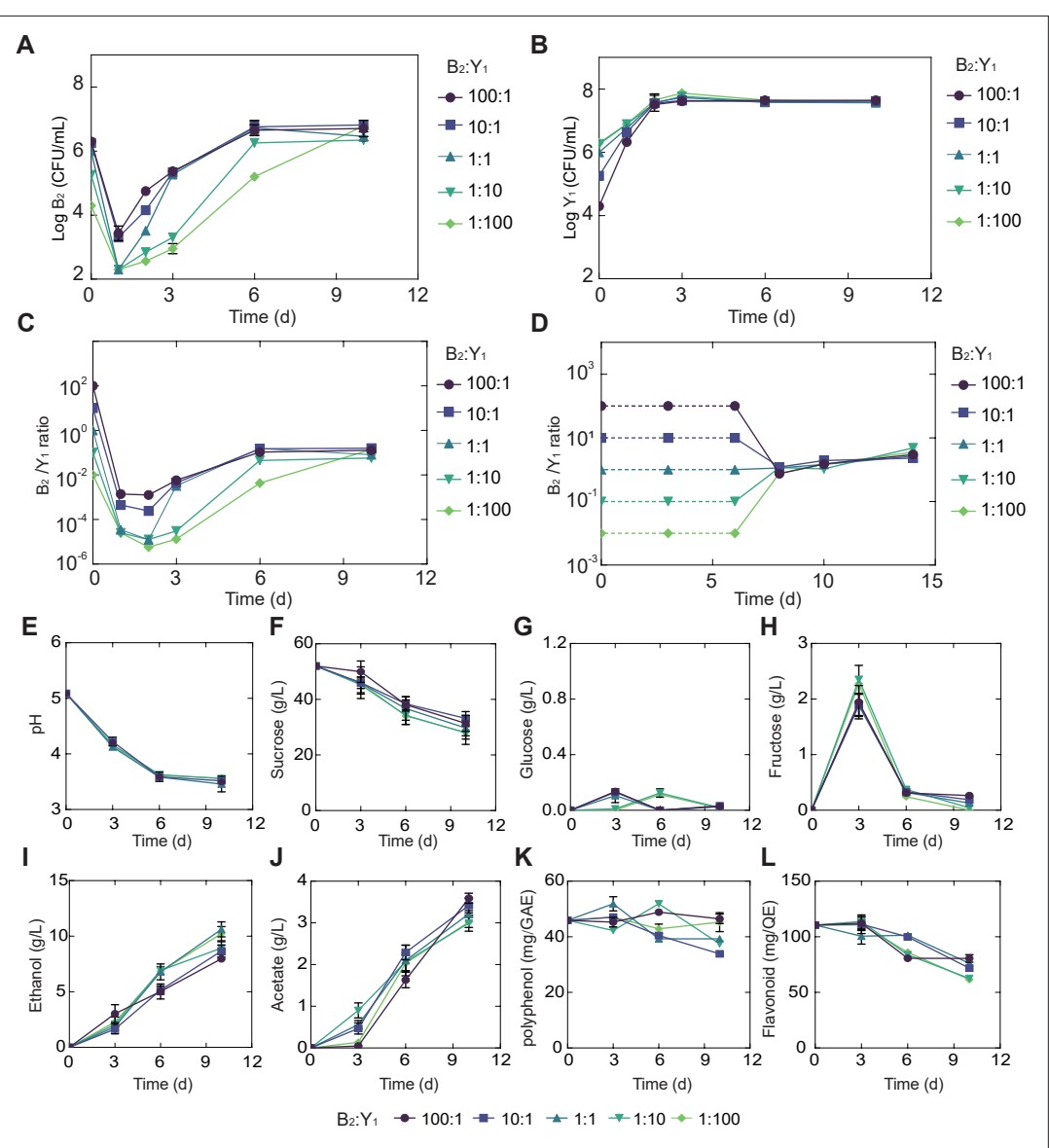

**Figure 3.** Temporal compositional and metabolic dynamics of the minimal core ($B_2Y_1$). (**A, B**) Populations of $B_2$ (**A**) and $Y_1$ (**B**) in broth throughout the course of a fermentation starting with 50 g/L sucrose. (**C**) Bacterium-to-yeast population ratio of the microbes in broth. (**D**) Ratio of microbial populations in pellicle during the fermentation. Notably, pellicle formation was not observed until day 6. (**E–L**) pH, carbon sources and metabolites during the fermentation driven by the core. The concentrations of total polyphenols and total flavonoids were expressed as mg of gallic acid equivalent (GAE) per mL of kombucha tea (mg GAE/mL) and mg of quercetin equivalent (QE) per mL of kombucha tea (mg QE/mL) respectively. Bars and error bars correspond to means and s.d.

The online version of this article includes the following source data and figure supplement(s) for figure 3:

**Source data 1.** Counting data of the minimal core.

**Source data 2.** Chemical property analysis of the minimal core.

**Figure supplement 1.** Population dynamics and pellicle formation of the minimal core ($B_2Y_1$).

Combining all three aspects of consideration, we chose $B_2Y_1$ as our minimal core of the KT microbiome for systematic, mechanistic investigation. Notably, $Y_1$ (*B. bruxellensis*) was also the most predominant yeast species in the native samples (*Figure 1C*, *Figure 2—source data 1*).

## Compositional and metabolic dynamics of the core

To reveal the detailed traits of the selected core ($B_2Y_1$), we performed a set of fermentation experiments with different initial ratios (100:1, 10:1, 1:1, 1:10, and 1:100) while maintaining a constant total inoculation ($2*10^6$ CFU/mL; Materials and methods). For all initial conditions, we found the bacterium $B_2$ decreased in day 1 but increased afterwards with a declining magnitude of the growth rate (*Figure 3A*, *Figure 3—figure supplement 1A*, *Figure 3—source data 1*). By contrast, the yeast $Y_1$ monotonically grew up with its rate reducing to null over time (*Figure 3B*, *Figure 3—figure supplement 1B*, *Figure 3—source data 1*). The population ratio of the two species showed that the community composition converged throughout the course of fermentation despite the variation of its initial ratio (*Figure 3C*).

The fermentation was also accompanied with the formation of pellicles (*Figure 3—figure supplement 1C*), which became visible after day 6 and grew continuously afterwards. Our CFU counting showed that, once pellicle formed, $B_2$ and $Y_1$ population densities remained relatively stable in the pellicles regardless of their initial abundance (*Figure 3—figure supplement 1D and E*). Meanwhile, their ratio converged to a fixed value (*Figure 3D*), although the dry weight of the pellicles increased over time (*Figure 3—figure supplement 1F*). The convergence of composition in both broth and pellicle suggested that there were underlying forces that drove and stabilized community population dynamics.

Additionally, we quantified the temporal biochemical characteristics of the KT broth. Strikingly, although initial population ratios were varied across four orders of magnitude, each of the variables including pH, sugars, acids and tea-derived chemicals converged onto its own consensus pattern (*Figure 3E–L*, *Figure 3—source data 2*), akin to the convergence of composition in broth and pellicle. Specifically, regardless of the initial population composition, the pH dropped from 5.0 to 3.5 through fermentation (*Figure 3E*), which was associated with continuous sucrose reduction (*Figure 3F*). Throughout the process, glucose remained at a low level (~0.1 g/L) (*Figure 3G*) while fructose was relatively higher with a pulse-like profile (*Figure 3H*). Acetate and ethanol on the other hand continued to accumulate during the fermentation (*Figure 3I and J*). Polyphenol and flavonoids remained relatively stable with minor decrease (*Figure 3K and L*). In the meanwhile, we found that throughout the fermentation process, the temporal kinetics of different metabolites were coordinated. For example, continuous pH reduction (*Figure 3E*) was in concert with sucrose drop (*Figure 3F*), which was anti-correlated with the increase of ethanol (*Figure 3I*) and acetate (*Figure 3J*). The observation of these patterns, such as the fructose spike, motivated us to investigate the underlying driving forces.

## Controlled fermentation assays yield causal claims for the core

To decode the mechanistic origins of the observed patterns, we investigated the metabolic processes of the constituting species ($B_2$ and $Y_1$) by conducting comprehensive monoculture fermentations with defined settings. Here, we focused on sucrose, glucose, fructose, ethanol and acetate as the primary biochemical substances of interest based on our measure of the KT broth and previous literature reports (*Jayabalan et al., 2014*; *Villarreal-Soto et al., 2018*). We used them alone and in combination as substrates to grow monocultures (Materials and methods) and quantified the temporal profiles of key substances, pH, biomass growth and pellicle formation, resulting a total of 30 panels (*Figure 4*, *Figure 4—source data 1*).

We harnessed the results of these panels to deduce biochemical conversion. As the starting carbon source, sucrose alone was not degradable by $B_2$ as shown in panel 1 (abbreviated as P1) of *Figure 4* but consumable by $Y_1$ with the production of a trace amount of glucose and fructose, ethanol accumulation, pH reduction and biomass growth (P16). Sucrose also showed weak hydrolysis in the presence of ethanol or acetate, which increased microbial survival (P6,7). Glucose and fructose were produced from sucrose hydrolysis primarily by $Y_1$ (16) and minorly by ethanol and acetate (P6,7). Glucose was efficiently utilized by $B_2$ for growth (P2) and by $Y_1$ with biomass and ethanol accumulation (P17). Fructose was consumable for $Y_1$ (P18), not $B_2$ (P3), with ethanol and biomass production. Fructose was also slowly converted to glucose in the presence of ethanol, which was mediated by $B_2$ probably

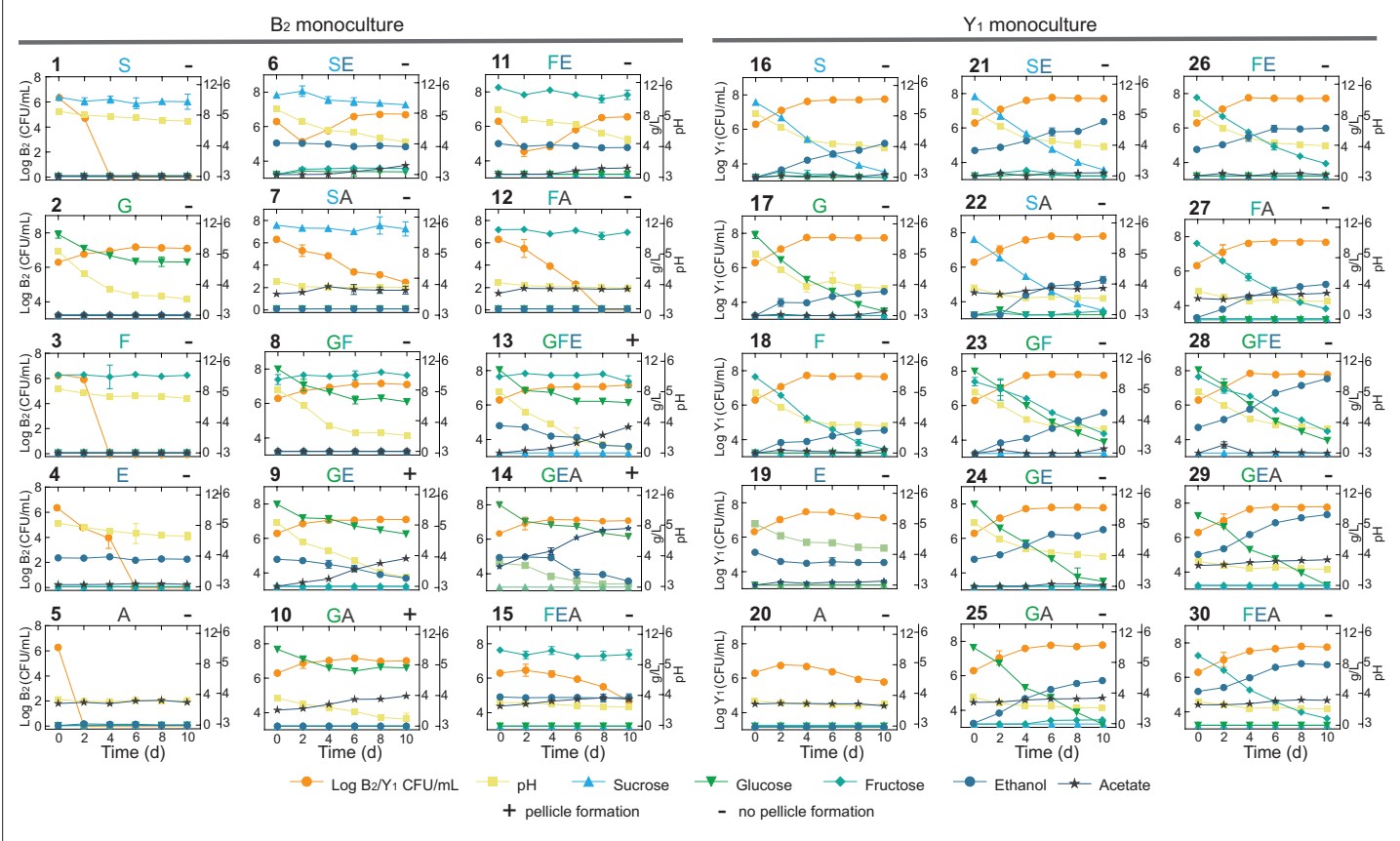

**Figure 4.** Comprehensive fermentation tests for the B₂ and Y₁ monocultures with different carbon sources. Sucrose (abbreviated as S, 10 g/L), glucose (G, 10 g/L), fructose (F, 10 g/L), ethanol (E, 50 mL/L), and acetate (A, 2 g/L) were used alone or in combination for fermentation. The number on top left of each panel is the label of the experiment. The letters on top middle of each panel indicate specific carbon sources used in the corresponding experiment. The +or – sign on the top right indicates whether a pellicle was formed during the fermentation. Bars and error bars correspond to means and s.d.

The online version of this article includes the following source data and figure supplement(s) for figure 4:

**Source data 1.** Counting and metabolic data of the B₂ and Y₁ monocultures.

**Figure supplement 1.** Images of pellicles formed by B₂ monoculture with different carbon sources.

and supported $B_2$ growth (P11). Ethanol was produced solely by $Y_1$ during the metabolism of sucrose, glucose and fructose (P16,17,18), not by $B_2$. Although ethanol alone was unusable by $B_2$ (P4), it was consumed with glucose (P9), resulting in acetate production and pellicle formation without obvious growth benefits compared to glucose alone. It thus implied that ethanol was used an energy source for pellicle formation as previously reported (*Molina-Ramírez et al., 2018*). Ethanol was also utilized by $Y_1$ in a weak fashion to result in biomass and acetate production (P19). Acetate was produced primarily by $B_2$ in the presence of multiple substrates (P6,9,13–15), particularly when glucose and ethanol were co-present (P9,13,14). In addition to $B_2$, $Y_1$ yielded a small amount of acetate with the consumption of sucrose, glucose, fructose or ethanol (P16-19). Acetate was additionally shown to minorly promote its own production by $B_2$ (P10 vs. P2) and ethanol production by $Y_1$ (P29,30 vs. P24,26).

Using the fermentation assays, we also inferred cellular tolerance to environmental stress. Comparison of the $B_2$ and $Y_1$ growth dynamics in single substrates showed that the yeast was more resistant than the bacterium to chemicals including ethanol and acetate (P4,5,19,20), which is another key factor that shapes community composition and metabolism. Additionally, the assays provided insight into pellicle formation. $B_2$ monoculture was capable of pellicle production (P9,10,13,14) (*Figure 4—figure supplement 1*), whereas $Y_1$ was deficient under all conditions (P16-30). Moreover, comparison of the pellicle-forming conditions (P9,10,13,14) with single substrate conditions (P1-5) showed that efficient biofilm development required not only glucose but also ethanol or acetate as a co-substrate.

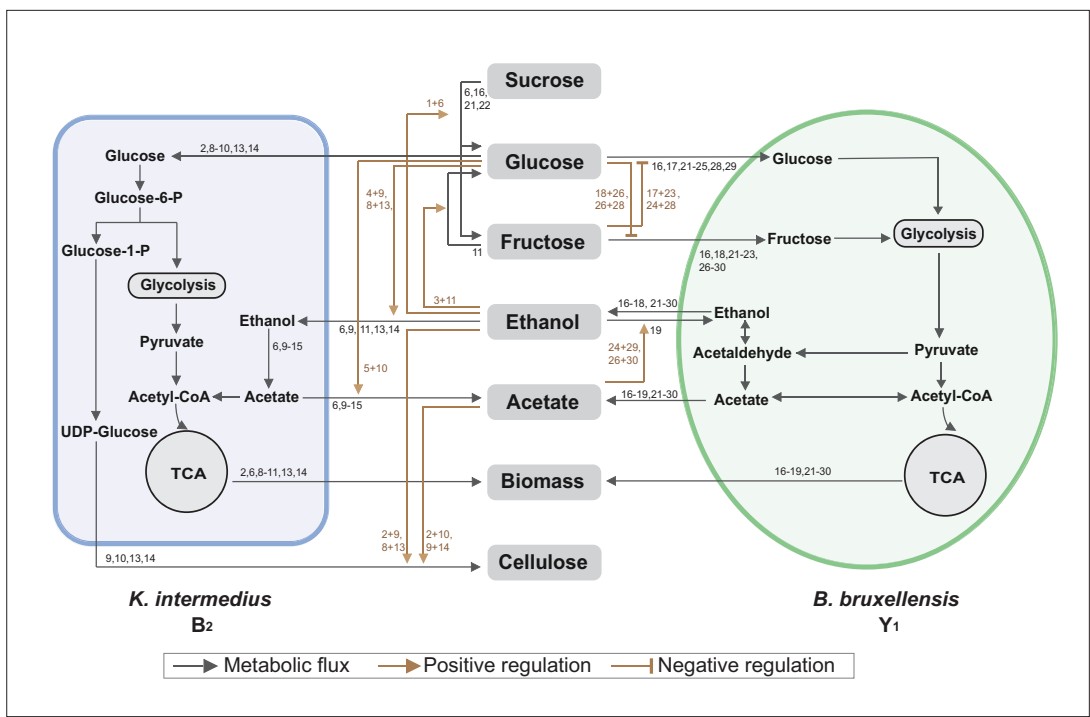

**Figure 5.** Summary of metabolic processes underlying the core. Black arrows refer to metabolic fluxes while brown arrows correspond to positive or negative regulatory interactions. The numbers associated with each arrow are the corresponding fermentation assays in *Figure 4* supporting the specific interaction.

The online version of this article includes the following figure supplement(s) for figure 5:

**Figure supplement 1.** Sucrose invertase activity of $Y_1$ monoculture and $B_2Y_1$ co-culture at different fermentation times.

**Figure supplement 2.** Comparison of the population and metabolic dynamics of $Y_1$ monoculture and $B_2Y_1$ co-culture.

**Figure supplement 3.** Population dynamics and metabolic profiles of the fermentations involving fixed $Y_1$ and varied $B_2$ initial abundances.

**Figure supplement 4.** Population dynamics and metabolic profiles of the fermentations involving fixed $B_2$ and varied $Y_1$ initial abundances.

Notably, although pellicle formation could occur in the presence of glucose as a sole carbon and energy source (*Masaoka et al., 1993*) for certain species, at least for those we investigated, it required two substrates to produce pellicle.

The above findings were synthesized and subsequently integrated with reported metabolic reactions (*Koschwanez et al., 2011*; *DeRisi et al., 1997*; *Smith and Divol, 2016*; *Liu et al., 2018*) into a system-level diagram (*Figure 5*), which involves major metabolic flows within each species, interspecies fluxes mediated by the environment, and regulatory effects from metabolites to fluxes. To illustrate its implications, we attempted to account for the observed compositional characteristics of the core. As the diagram showed, $Y_1$ breaks down sucrose into glucose and fructose for its own growth, which also benefits $B_2$ by sharing glucose. Additionally, $Y_1$ secretes ethanol that is utilized by $B_2$ when glucose is present. Thus, the core possesses a commensal relationship whereby $Y_1$ provides two modes of benefits to $B_2$. By design, such an interaction confers the stability and convergence of the ecosystem composition, thus providing a mechanistic driver for the population convergence in broth and pellicle (*Figure 3C and D*) and echoing with previous work demonstrating that interpopulation cooperation leads to stable composition (*Momeni et al., 2013*).

The results also elucidated three ways in which $Y_1$ is more robust than $B_2$: first, $B_2$ relies on $Y_1$ for glucose release; second, $Y_1$ is more versatile for utilizing different substrates including sucrose, glucose, fructose, and ethanol; third, $Y_1$ has a higher tolerance to ethanol and acetate. These findings

explained the temporal growth difference that $B_2$ declined first before recovery while $Y_1$ monotonically grew since the beginning of fermentation (*Figure 3A, B*).

Toward metabolic characteristics, the population of each species is a key determinant because total extracellular metabolites are determined by the productivity of individual cells multiplied by cell populations. Thus, for the same set of microbial species, the commensal interaction-which caused the convergence of population composition-also drove the convergence of temporal profiles of substrates, pH and metabolites as shown in *Figure 3E-L*. Additionally, although glucose and fructose were hydrolyzed simultaneously from sucrose, the former was consumed by both $B_2$ and $Y_1$ while the latter was useable exclusively for $Y_1$, which resulted in a constant low level of glucose but a relatively higher level of fructose (*Figure 3G, H*). Moreover, utilizations of glucose and fructose were accompanied with the release of ethanol and the both sugars were derived from sucrose hydrolysis; thus, ethanol increase was anti-correlated with sucrose decrease (*Figure 3F, I*). Acetate was mainly produced by $B_2$ in the presence of glucose and ethanol, both of which were converted directly or indirectly from sucrose; therefore, acetate accumulation was positively associated with sucrose consumption (*Figure 3F, J*).

For pellicle formation, the diagram showed that $B_2$ was solely responsible for pellicle formation. Meanwhile, it was $Y_1$ that provided glucose and ethanol needed by $B_2$. Such a cooperative relationship accounted for the findings that $B_2$ or $Y_1$ alone was deficient in pellicle formation, and it needed the co-culture instead.

Relating to the bacterium-yeast symbiosis, some previous studies reported that the microbial social interactions are commensal while others concluded to be mutual (*Laavanya et al., 2021*; *May et al., 2019*). To resolve this debate, we conducted experiments to examine possible benefits from $B_2$ to $Y_1$. As certain yeasts were suggested to secret more invertase when co-cultured with cheaters (*Celiker and Gore, 2012*), we measured the invertase activity of $Y_1$ in monoculture and in co-culture with $B_2$ but did not find significant difference between the two conditions ($p < 0.05$; *Figure 5—figure supplement 1*). We also compared the growth and metabolites of $Y_1$ in the $B_2Y_1$ co-culture and in monoculture with different initial ratios; however, the results showed that $B_2$ did not affect either growth or metabolites except the increase of acetate which was produced by $B_2$ (*Figure 5—figure supplement 2*). We additionally varied the $B_2$ level while fixing $Y_1$'s initial amount and altered $Y_1$ while maintaining the initial $B_2$. In both settings, $Y_1$ growth was not affected by $B_2$, and all metabolic variables, except acetate, exhibited the same patterns (*Figure 5—figure supplements 3 and 4*). Therefore, although we did not rule out the possibility of altered interactions across SCOBYs, our experiments demonstrated that, at least in our system, the symbiosis driving the community is commensal instead of mutual.

## Insights into communities with increased complexity and varied conditions

We have thus far illustrated the casual claims for the two-species core, but do these findings provide implications for communities with different complexity and settings? To answer the question, we assembled a consortium of 10 species ($B_1$, $B_2$, $B_3$, $B_4$, $B_5$, $Y_1$, $Y_2$, $Y_3$, $Y_4$, and $Y_5$). Using the consortium, we performed fermentations with the same medium as previous (i.e. black tea substrate supplemented with 50 g/L sucrose) and using different initial total bacteria-to-yeast ratios while keeping all bacterial species even and all yeast species even (*Figure 6A*).

In bulk, the ten-species community yielded the same patterns as the two-species core, including the overall compositional convergence compared to the initial structure (*Figure 6B*, *Figure 6—source data 1*), continued sucrose consumption (*Figure 6C*, *Figure 6—source data 1*), increase in ethanol and acetate (*Figure 6D, E*), monotonic pH reduction (*Figure 6—figure supplement 1B*), consistent lowness of glucose (*Figure 6—figure supplement 1C*), pulse-like fructose profile (*Figure 6—figure supplement 1D*) and successful pellicle formation (data not shown). Although species composition was not directly analyzed, the similarity in patterns suggested that the core served as a good approximation of the ten-species consortium and that the knowledge from the simple core provided predictive insights into the functions of communities with an increased degree of complexity.

Meanwhile, in detail, the two ecosystems showed differences in specific profiles. In the ten-species community, the composition converged from days 0 to 6 but diverged at day 10 (*Figure 6B*), different from the continuous convergence of the core (*Figure 3C, D*). Compared to the core (*Figure 3*), the ten-species community also yielded different metabolic patterns: its pH dropped faster (*Figure 6—figure supplement 1B*), sucrose was consumed quicker (*Figure 6C*), fructose was more sensitive to

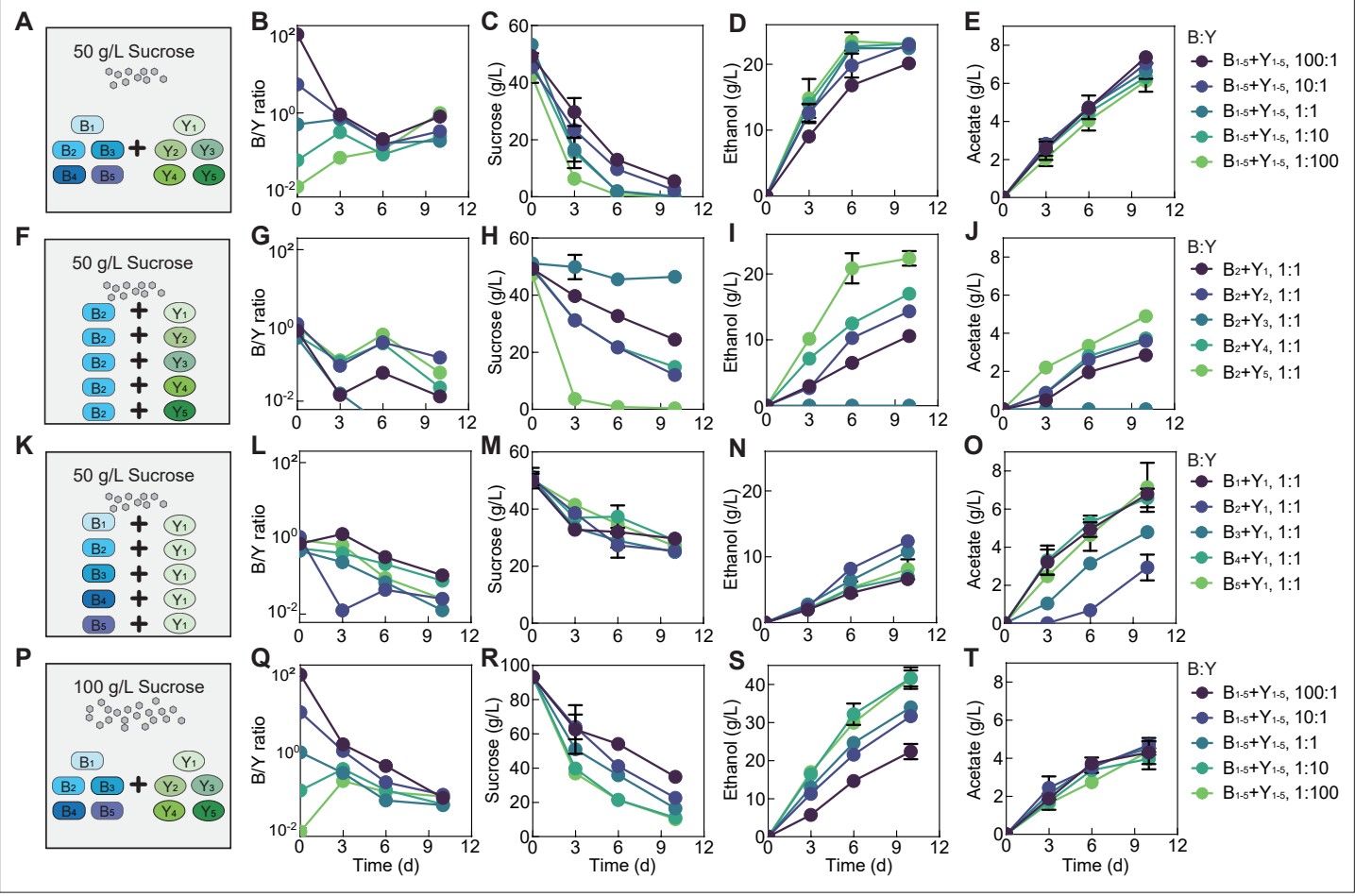

**Figure 6.** Fermentation by synthetic communities with increased complexity and altered conditions. (**A**) Schematic illustration of a ten-species community involving $B_1$-$B_5$ and $Y_1$-$Y_5$ in a fermentation with 50 g/L of initial sucrose. (**B–E**) Population ratio (**B**), sucrose (**C**), ethanol (**D**), and acetate (**E**) throughout the course of the fermentation shown in (**A**). (**F**) Schematic illustration of 5 two-species communities with each involving $B_2$ and one of the yeasts ($Y_1$-$Y_5$) in a fermentation starting with 50 g/L sucrose. (**G–J**) Population ratio, sucrose, ethanol, and acetate during the fermentation illustrated in (**F**). (**K**) Schematic illustration of five two-species communities with each involving $Y_1$ and one of the bacteria ($B_1$-$B_5$) in a fermentation with 50 g/L of initial sucrose. (**L–O**) Population ratio, sucrose, ethanol and acetate during the course of fermentation illustrated in (**K**). (**P**) Schematic illustration of the ten-species community involving $B_1$-$B_5$ and $Y_1$-$Y_5$ in a fermentation starting with 100 g/L sucrose. (**Q–T**) Population ratio, sucrose, ethanol and acetate during the fermentation depicted in (**P**). Bars and error bars correspond to means and s.d.

The online version of this article includes the following source data and figure supplement(s) for figure 6:

**Source data 1.** Counting and metabolic data of the ten-species communities.

**Source data 2.** Counting and metabolic data of the two-species communities.

**Figure supplement 1.** Fermentation by synthetic communities with increased complexity and altered conditions.

**Figure supplement 2.** Temporal compositional and metabolic dynamics of the minimal core ($B_2Y_1$) during a fermentation with 5 g/L of initial sucrose.

**Figure supplement 3.** Temporal compositional and metabolic dynamics of the minimal core ($B_2Y_1$) during a fermentation with 100 g/L of initial sucrose.

initial conditions (*Figure 6—figure supplement 1D*), ethanol accumulated faster and nonlinearly with rapid production from days 0 to 6 followed by slower increase or cease from days 6 to 10 (*Figure 6D*), and acetate increased faster (*Figure 6E*).

We speculated that these differences arose from the variability of the metabolic capacities of members constituting the communities. To test the speculation, we repeated the fermentation with five two-species co-cultures involving $B_2$ and different yeasts (*Figure 6F–J*, *Figure 6—figure supplement 1E-H*). The results confirmed that the yeasts were highly variable in sucrose consumption with $Y_5$ being the strongest and $Y_3$ the weakest (*Figure 6H*). Additionally, owing to the coordinated metabolism revealed via the core, rapid sucrose consumption by $B_2Y_5$ was associated with a relatively high

$B_2$ abundance, a large pulse of glucose and fructose, rapid ethanol and acetate production and quick pH drop; by contrast, weak sucrose consumption by $B_2Y_3$ were accompanied with a low $B_2$ abundance, an undetectable level of glucose and fructose, abolished ethanol and acetate production, and slow pH reduction (*Figure 6G–J*, *Figure 6—figure supplement 1F-H*). We also performed fermentation using the co-cultures of $Y_1$ with different bacterial species (*Figure 6K–O*, *Figure 6—figure supplement 1I-L*). The five ecosystems showed a comparable sucrose consumption rate (*Figure 6M*), suggesting that sucrose degradation was dictated primarily by yeast species although certain bacterial species (e.g. $B_1$, $B_3$, and $B_5$ in *Figure 2A*) could contribute. Meanwhile, the fermentations yielded varied ethanol and acetate patterns which were anti-correlated (*Figure 6N and O*), suggesting that the ethanol-to-acetate conversion of the bacterial species were variable with $B_1$ being the strongest and $B_2$ the weakest.

From the above experiments, a mechanistic origin underlying the compositional and metabolic differences of the two- and ten-species communities emerged as following. Compared to the two-species core, the 10-species community had a higher overall sucrose consumption rate and a higher ethanol-to-acetate conversion rate which were averaged from the rates of the involved yeasts and bacteria. As a result, the ten-species community consumed sucrose faster (*Figure 6C*), which subsequently led to a higher level of fructose, ethanol and acetate as well as a faster pH reduction (*Figure 6D, E*, *Figure 6—figure supplement 1B, D*). Meanwhile, rapid sucrose degradation resulted in sucrose depletion in the middle of fermentation when the fermentation started with a high relative yeast abundance (e.g. 1:1, 1:10, 1:100) (*Figure 6C*), which forced the yeast to metabolize fructose and ethanol instead of producing them. Under these scenarios, ethanol had a nonlinear pattern with rapid accumulation in the first few days and a slow increase or cease from days 6 to 10 (*Figure 6D*). Meanwhile, as the commensal bacteria-yeast interaction relied primarily on the yeast to breakdown sucrose to provide glucose and ethanol for the bacteria, sucrose depletion also altered the strength of the symbiosis, which consequently shaped the dynamics of population convergence (*Figure 6B*) because community dynamics was driven by the symbiosis.

Based on the finding that sucrose depletion shifted compositional and metabolic patterns, we hypothesized that, for the 10-species community, increasing sucrose availability could prevent sucrose depletion and hence drive its patterns closer to those of the core. We tested the hypothesis by performing the fermentation with 100 g/L sucrose (*Figure 6P–T*, *Figure 6—figure supplement 1M-P*). Indeed, the results showed that the microbial composition continued to converge throughout the course of fermentation (*Figure 6Q*) instead of first convergence then divergence in *Figure 6B*. Meanwhile, the continuous sucrose reduction (*Figure 6R*) was accompanied with a faster and approximately linear ethanol increase (*Figure 6S*), a lower rate of acetate accumulation (*Figure 6T*), and a higher level of glucose and fructose (*Figure 6—figure supplement 1O, P*) compared to the 50 g/L sucrose case (*Figure 6B–D*, *Figure 6—figure supplement 1B-D*). Notably, here the glucose and fructose patterns (*Figure 6—figure supplement 1O, P*) were still different from those of the core (*Figure 3G, H*) because the ten-species community was much more efficient than the core for sucrose hydrolysis.

We further reasoned that the dependence of ecosystem characteristics on sucrose availability was not unique to the ten-species community and shall also apply to the two-species core. To test the reasoning, we performed the fermentation with the core using 5 g/L of sucrose (*Figure 6—figure supplement 2*). Remarkably, the composition converged in the first 3 days but diverged afterwards (*Figure 6—figure supplement 2A*) and ethanol started with linear increase initially but declined after day 3 (*Figure 6—figure supplement 2H*), similar to the case of the ten-species community with 50 g/L sucrose (*Figure 6B, D*). Conversely, when we increased the initial sucrose concentration to 100 g/L (*Figure 6—figure supplement 3*), the compositional convergence of the core was restored (*Figure 6—figure supplement 3A*) and the ethanol profile became continuous accumulation (*Figure 6—figure supplement 3H*).

The results from the core and the 10-species community both informed that an increase in sucrose consumption results in a reduction or depletion in sucrose, an augmentation in the production of glucose, fructose, ethanol and acetate along with a reduction in environmental pH. Interestingly, such a relationship also explained the seemingly abnormal metabolite patterns of sample C, the outlier of the four native KT microbiome samples, that we observed at the beginning of our study (*Figure 1D*). In that regard, our findings provided the mechanistic basis to understand the variations of metabolite patterns among the original microbiome samples.

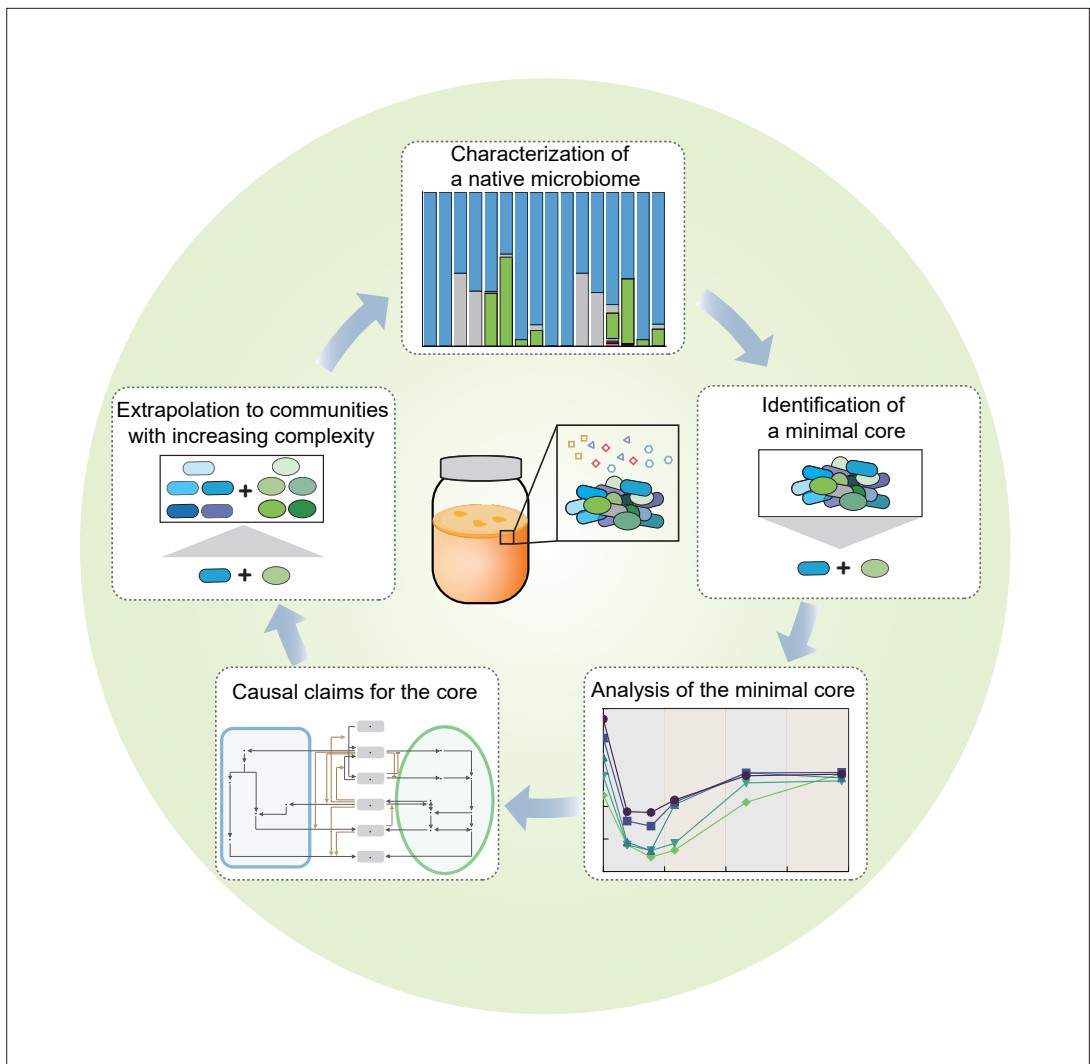

**Figure 7.** Summary of the overall concept and steps of our complexity-reduction approach for microbiome dissection.

Together, our experiments demonstrated that the knowledge from the minimal core offered predictive bulky insights into the traits of communities with varied system complexity and fermentation conditions. Meanwhile, the diversity in metabolic capacities, which was caused by the increase in species richness, accounted for the differences between the patterns of complex and minimal communities.

## Discussion

With rapid advances in species cataloging and correlation analysis, one remaining grand challenge in microbiome research is to uncover causal claims that dictate microbial composition and function (*Fischbach, 2018*; *Gilbert et al., 2018*; *Schmidt et al., 2018*; *Skelly et al., 2019*; *Taroncher-Oldenburg et al., 2018*; *Lv et al., 2021*). In this work, we present the identification, characterization, and utilization of a minimal core for elucidating the molecular mechanisms driving the KT microbiome. We showed that metabolic underpinnings specified the structural and metabolic characteristics of the core and also provided insights into the behaviors of communities with increased complexity and altered conditions.

*Figure 7* summarizes the overall concept and steps of our complexity-reduction approach to dissect microbiome. Lying at the heart of our study is the reduction in system complexity, which

involves three key steps: identification of a core simplified from a native microbiome but capable of resembling the native, characterization of metabolic underpinnings of the core, and extrapolation of the knowledge from the core to communities with altered complexity and conditions. Such a strategy is conceptually analogous to the motif-based study of complex networks whereby core functional motifs, such as feedback loops and feed-forward regulation, are repeatedly identified as building blocks of networks and utilized to understand complex system behaviors (*Alon, 2007*). Although this study focused exclusively on the KT microbiome, the strategy demonstrated here is not limited to the specific ecosystem. Given its systematic nature, we expect that it may be extendable to other microbial communities. In that regard, our strategy provides a promising solution to address system complexity, a major hurdle for mechanistic investigation of microbiome.

Notably, although minimal cores serve as attractive alternatives to complex ecosystems, they are not intended to substitute native microbiomes. With the increase of complexity, certain compositional and metabolic traits identified in a core may be altered in its native counterpart. Conversely, novel properties may emerge when species richness increases. Thus, minimal cores provide a point of entry to unlock the mechanistic behaviors of a community, which shall be followed by the analysis of the full system for systematic understanding. Meanwhile, defining a proper core is critical for successful implementation of our framework. In this study, we successfully identified and utilized a two-species consortium as the core for systematic analysis of the KT microbiome. However, such a strategy may become less straightforward for many native microbiomes, such as those in the soil and in the human gut, due to their remarkable degree of complexity and their close association with the corresponding hosts (e.g., plant and human). Additionally, the KT microbiome studied here was driven primarily by pairwise interactions, another factor that made it possible to use the two-species ecosystem as an approximate of the entire community. For those involving higher order interactions (*Billick and Case, 1994*; *Grilli et al., 2017*; *Liu et al., 2020*), microbial social interactions are much more complicated, and it needs significantly more efforts to screen and determine valid cores. In these cases, whether and to what extent a relatively simple core can be defined will depend on the specific starting microbiome, as well as the target functions of interest. In addition, a single microbiome in principle can possess multiple cores depending on different selection criteria, such as abundance, temporal pattern and function. Future efforts in these aspects are needed to fully realize the power of this community analysis strategy.

A major goal of the food industry is to improve food quality and flavor through the optimization of starter culture and fermentation process (*Vinicius De Melo Pereira et al., 2019*; *Zhao et al., 2019*; *Jin et al., 2017*). The synthetic core developed here successfully drove the KT fermentation, thereby serving as an effective, functional culture starter. Compared to the native microbiome, this well-defined, synthetic system offers a controllable platform to modulate the starter composition and metabolite secretion during fermentation. Therefore, the work also gives a potential solution to systematically tailor fermented foods with desired traits.

# Materials and methods

## Key resources table

| Reagent type (species) or resource | Designation | Source or reference | Identifiers | Additional information |
|---|---|---|---|---|
| Commercial assay or kit | Fecal/soil Microbe Miniprep kit | ZYMO | Cat No./ID:D6010 | |
| Commercial assay or kit | Gluconic acid Kit | Megazyme, Ireland | Cat No./ID:K-GATE | |
| Sequence-based reagent | B-f | *Huang et al., 2021* | PCR Primer | Forward primer used for amplifying bacterial DNA for Sanger sequencing |
| Sequence-based reagent | B-r | *Huang et al., 2021* | PCR Primer | Reverse primer used for amplifying bacterial DNA for Sanger sequencing |
| Sequence-based reagent | NL-1 | *Coton et al., 2017* | PCR Primer | Forward primer used for amplifying yeast DNA for Sanger sequencing |
| Sequence-based reagent | NL-4 | *Coton et al., 2017* | PCR Primer | Reverse primer used for amplifying yeast DNA for Sanger sequencing |

*Continued on next page*

*Continued*

| Reagent type (species) or resource | Designation | Source or reference | Identifiers | Additional information |
|---|---|---|---|---|
| Software, algorithm | Canoco | Microcomputer Power, Ithaca, NY | | Version 5.0 |
| Software, algorithm | SIMCA | Umetricus, Sweden | | Version 14.1 |

## Kombucha tea fermentation

Black tea (Harney & Sons Fine Teas, Millerton, NY) was purchased as the tea substrate for the fermentation. The live starter culture SCOBY used as inoculum were obtained from 4 different commercial sources, referring to samples A, B, C, and D. Kombucha tea was prepared as previously reported with minor modifications (*Jayabalan et al., 2007*; *Cardoso et al., 2020*). Briefly, 1 L deionized water was boiled, added with 12 g/L black tea and allowed to infuse for 5 min. After removing the tea leaves, sucrose (50 g/L) was dissolved in hot tea. After cooling, the tea mixture was filtered through sterile sieve to 500 mL glass vessel with cotton and gauze caps. Then, 3.0% SCOBY and liquid broth (10% v/v) of the SCOBY samples were added to tea broth. The kombucha tea was incubated at 25 °C for 14 days.

## Amplicon sequencing of 16S ribosomal RNA (rRNA) and ITSs

Pellicle samples were first treated with 200 mg/mL cellulase (Sigma-Aldrich, Milan, Italy) for 16 hr, and sonicated in ice bath for 1 min using a probe sonicator (Model 505, Fisherbrand, USA) for 1 min. Then the samples were centrifuged at 6500 rpm at 4 °C for 10 min. The cell pellets were used for DNA extraction. Total DNA extractions were performed for tea broth and pellicle samples using Quick-DNA Fecal/soil Microbe Miniprep kit (ZYMO Research Corp.) according to the manufacturer's instructions.

16 S rRNA gene and ITS amplicon sequencing library constructions and Illumina MiSeq sequencing were conducted by GENEWIZ, Inc (South Plainfield, NJ, USA). Sequencing library was prepared using a MetaVx 16 s rDNA Library Preparation kit and ITS-2 Library Preparation kit (GENEWIZ, Inc, South Plainfield, NJ, USA). Briefly, for each sample, 50 ng DNA was used to generate amplicons that cover the V3 and V4 hypervariable regions of bacteria and ITS-2 hypervariable region of fungi. Afterwards, each sample was added with indexed adapters. The barcoded amplicons were sequenced on the Illumina MiSeq platform using 2×250 paired-end (PE) configuration (Illumina, San Diego, CA, USA).

Raw sequence data was converted into FASTQ files and de-multiplexed using Illumina's bcl2fastq 2.17 software. QIIME data analysis package was used for 16 S rRNA and ITS rRNA data analysis (*Caporaso et al., 2010*). All the reads (forward and reverse) were assigned to different samples based on barcode, and then truncated by cutting off the primer and barcode. After quality filtering (*Ren et al., 2019*), the sequences were compared with the RDP Gold database to detect chimeric sequences using the UCHIME algorithm (*Edgar et al., 2011*). Subsequently, the effective sequences were grouped into operational taxonomic units (OTUs) using the clustering program VSEARCH (1.9.6) against the Silva 119 database for bacteria (*Quast et al., 2013*) and the UNITE ITS database for fungi (*Nilsson et al., 2019*), with pre-clustered at 97% of sequence identity. The Ribosomal Database Program (RDP) classifier was used to assign taxonomic category to all OTUs at a confidence threshold of 80% (*Cole et al., 2009*).

## Species isolation and identification

For species isolation, kombucha tea broths were diluted and plated directly whereas pellicle samples were sonicated and digested before dilution as described above. Diluted samples were then inoculated in different selective media. For bacterial isolation, de Man, Rogosa and Sharpe medium (MRS), Mannitol medium (*Coton et al., 2017*), and Glucose yeast extract calcium carbonate medium (GYC) (*Kim et al., 2019*) were used in conjugation with 0.1% cycloheximide or 500 μg/mL natamycin for inhibiting fungi growth. Isolation of yeast species was carried out using the yeast extract peptone dextrose (YPD) medium supplemented with 100 mg/L chloramphenicol. Isolated species were identified by Sanger sequencing of the 16 S and 26 S rRNA gene regions, with the universal primers B-f (5'-AGAGTTTAGTCCTGGCTCAG-3') and B-r (5'- AAGGAGGTGATCCAGCCGCA-3') for bacteria

(*Huang et al., 2021*), and NL-1 (5´-GCATATCAATAAGCGGAGGAAAAG-3´) and NL-4 (5´-GGTC-CGTGTTTCAAGACGG-3´) for yeasts (*Coton et al., 2017*).

## Biochemical analyses

The pH was measured with a pH meter (AE150; Fisher Scientific, Waltham, MA) inserted directly into samples. Acetate, glucuronate and ethanol concentrations were determined by high performance liquid chromatography (HPLC, Agilent Technologies 1,200 Series) equipped with a refractive index detector using a Rezex ROA Organic Acid H+ (8%) column (Phenomenex Inc Germany). The column was eluted with 0.005 N of $H_2SO_4$ at a flow rate of 0.6 mL/min at 50 °C (*Ha et al., 2011*). Sucrose, glucose and fructose were analyzed using RCM Monosaccharide $Ca^{2+}$ column (Phenomenex Inc, Germany). The column was eluted with deionized water at a flow rate of 0.6 mL/min at 80°C (*Ilaslan et al., 2015*). For gluconate detection, the gluconic acid Kit (Megazyme, Ireland) was used. The concentration of total polyphenols was measured by the Folin-Ciocalteu colorimetric method, with gallic acid as standard. The absorbance was measured at 765 nm and the results were expressed as mg of gallic acid equivalent (GAE) per mL of kombucha tea (mg GAE/mL) (*Bhattacharya et al., 2013*). The total flavonoids were determined using an aluminum chloride assay using quercetin as standard. The absorbance was measured at 430 nm and the content was expressed as mg of quercetin equivalent (QE) per mL of kombucha tea (mg QE/mL) (*Sun et al., 2015*). The invertase activity was determined according to the method described by Laurent et al (*Laurent et al., 2020*). The remaining sucrose was detected by HPLC as described above. The measurement of pellicle weight was based on the descriptions of Florea et al. using 0.1 M NaOH for pretreatment (*Florea et al., 2016*).

## Co-culture fermentation experiments

All the stocked bacteria and yeasts isolates were grown in YPD media and then centrifuged and washed twice with fresh tea liquid (12 g/L) at 6500 g for 5 min. Synthetic, pairwise bacterium-yeast cocultures were assessed in tea liquid with 50 g/L sucrose. Bacteria species included *Komagataeibacter rhaeticus* ($B_1$), *Komagataeibacter intermedius* ($B_2$), *Gluconacetobacter europaeus* ($B_3$), *Gluconobacter oxydans* ($B_4$) and *Acetobacter senegalensis* ($B_5$). Yeasts included *Brettanomyces bruxellensis* ($Y_1$), *Zygosaccharomyces bailii* ($Y_2$), *Candida sake* ($Y_3$), *Lachancea fermentati* ($Y_4$) and *Schizosaccharomyces pombe* ($Y_5$). For each pairwise co-culture, the total inoculation was as a final amount at $2*10^6$ CFU/mL and the inoculation amounts of bacteria and yeast were equal. Monoculture of each species was used as control group and the inoculation was also as a final amount at $2*10^6$ CFU/mL. The cultures were then incubated at 30 °C, and microbial populations and biochemical parameters were measured after 10 days fermentation. To count bacteria and yeasts, 1000 µg/mL of natamycin or 100 mg/L chloramphenicol of was added respectively.

The $B_2$-$Y_1$ consortium was fermented in tea liquid supplemented with 5, 50, 100 g/L sucrose individually. To characterize the consortium, $B_2$ and $Y_1$ were inoculated at different initial ratios from 100:1 to 10:1, 1:1, 1:10, and 1:100. The growth rates of $B_2$ and $Y_1$ and the $B_2/Y_1$ ratio were calculated. Meanwhile, to determine the effect of $Y_1$ on $B_2$, we performed the $Y_1$ monoculture experiment using the same inoculation amount as the $B_2Y_1$ co-culture. Moreover, we fixed the inoculation of $B_2$ or $Y_1$ ($1*10^6$ CFU/mL) but varied the amount of the other species from 0 to $1*10^4$, $1*10^5$, $1*10^6$, $1*10^7$ CFU/mL. Additionally, to determine if different species differ in growth and metabolic ability, $Y_1$ was co-cultured with different bacterial species ($B_1$, $B_2$, $B_3$, $B_4$, and $B_5$) and $B_2$ was co-cultured with different fungal species ($Y_1$, $Y_2$, $Y_3$, $Y_4$ or $Y_5$) in tea substrate supplemented with 50 g/L sucrose. The population dynamics and biochemical parameters were measured at 0, 3, 6, 10 days or 0, 1, 2, 3, 6, 10 days. To count microbes in pellicles, the pellicles were first digested by shaking for 16 hr at 4 °C in 15 ml of PBS buffer with 2% cellulase (Sigma Aldrich, C2730).

## Monoculture fermentation with different carbon sources

To uncover the metabolic underpinnings that drive microbial population dynamics and metabolite synthesis, we conducted a series of monoculture growth experiments for $B_2$ and $Y_1$ using different carbon sources. Specifically, we used 10 g/L sucrose, 10 g/L fructose, 10 g/L glucose, 50 mg/L ethanol, and 2 g/L acetate for fermentation. The initial inoculation of $B_2$ and $Y_1$ was $2*10^6$ CFU/mL. The population and biochemical parameters were measured at 2 days intervals.

## Construction and fermentation of communities with increased complexity

The five bacterial isolates (B$_1$, B$_2$, B$_3$, B$_4$, and B$_5$) and the five yeast isolates (Y$_1$, Y$_2$, Y$_3$, Y$_4$, and Y$_5$) were pooled together to create a synthetic, ten-species community. In initial inoculations, all bacterial species were equally abundant, and all yeast species were also equal; however, the total bacteria-to-yeasts ratio was varied from 100:1, 10:1,1:1, 1:10, to 1:100 while fixing the total amount of inoculation (2*10$^6$ CFU/mL). Two different sucrose levels, 50 and 100 g/L, were added to tea liquid for fermentation. The population dynamics and biochemical parameters were measured at 0, 3, 6, 10 days.

## Statistical analysis

All the experiments were performed for three times. Redundancy analysis between microbial community and metabolites was performed with Canoco 5.0 software (Microcomputer Power, Ithaca, NY). The hierarchical cluster analysis and principal component analysis on different consortia were performed with the SIMCA-14.1 software (Umetricus, Sweden). For hierarchical cluster analysis, the distances between observations were calculated using Ward's method based on the concentrations of different metabolites. Heatmaps of the chemical properties of the 25 two-species fermentations and 10 single-species fermentations were produced using the heatmap package with Z-score normalization (*Quan et al., 2019*) in R.

## Additional information

### Funding

| Funder | Grant reference number | Author |
| --- | --- | --- |
| National Science Foundation | 1553649 | Xiaoning Huang Ting Lu |
| University of Illinois Urbana-Champaign | | Yongping Xin Ting Lu |

The funders had no role in study design, data collection and interpretation, or the decision to submit the work for publication.

### Author contributions

Xiaoning Huang, Data curation, Formal analysis, Investigation, Methodology, Validation, Visualization, Writing – original draft; Yongping Xin, Investigation, Methodology, Validation; Ting Lu, Conceptualization, Formal analysis, Funding acquisition, Project administration, Supervision, Writing – original draft, Writing – review and editing

### Author ORCIDs

Xiaoning Huang (iD) http://orcid.org/0000-0002-5858-6281
Ting Lu (iD) http://orcid.org/0000-0001-9043-3253

### Decision letter and Author response

Decision letter https://doi.org/10.7554/eLife.76401.sa1
Author response https://doi.org/10.7554/eLife.76401.sa2

## Additional files

### Supplementary files

• Transparent reporting form

### Data availability

Amplicon sequencing data are deposited at the NCBI and available under a Bioproject ID PRJNA764354. Reference sequences of all bacterial and yeasts isolates are deposited at the NCBI. Figure 2-source data 1 contains accession numbers for all of the sequences.

The following dataset was generated:

| Author(s) | Year | Dataset title | Dataset URL | Database and Identifier |
|---|---|---|---|---|
| Huang X, Xin Y, Lu T | 2021 | Kombucha tea microbiome | https://www.ncbi.nlm.nih.gov/search/all/?term=PRJNA764354 | NCBI Gene Expression Omnibus, PRJNA764354 |

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
