## [Editor Report]

This work will be of interest for researchers studying the functions of microbial communities, microbial ecology and interactions. Using the Kombucha tea (KT) microbiome as a case study, Huang et al., provide a framework for simplifying complex communities into core communities that capture aspects of complex communities. Authors demonstrated that core communities can facilitate a mechanistic understanding of how microbes interact, especially when member species are individually culturable. The work presents a fresh, novel approach for the coarse-grained analysis of complex microbiomes.

---

## [Decision Letter]

**Decision letter after peer review:**

Thank you for submitting your article "A Systematic, Complexity-Reduction Approach to Dissect Microbiome: the Kombucha Tea Microbiome as an Example" for consideration by *eLife*. Your article has been reviewed by 3 peer reviewers, including Wenying Shou as Reviewing Editor and Reviewer #1, and the evaluation has been overseen by Meredith Schuman as the Senior Editor.

Essential revisions:

Your revised submission must address these two points resulting from the consultative review process. For specific guidance on both points, please see the individual reviews. Other points brought up by the reviewers do not have to be addressed in your point-by-point response.

(1) Clearer, more informative figures and better explanations of figures.

(2) Discuss limitations of the approach (e.g. in vitro fermentation communities are easier as the host is not a factor; the original community is already quite simple compared to other types of communities).

*Reviewer #1 (Recommendations for the authors):*

I have several comments:

1) How can fructose be slowly converted to glucose in the presence of ethanol? I presume that this is mediated by B2?

2) Figures can be made more clear:

Figure 2b: the absolute concentrations need to be indicated to interpret the graph. Not clear what +3 and -3 mean.

Figure 3a and b: please plot dynamics (absolute abundance) in semilog plots. Rates can then be easily estimated from slopes.

Figure 4 is complex, and hard to read. Use contrasting colors, and colormatch with labels S/G/F/E/A.

*Reviewer #2 (Recommendations for the authors):*

The paper is well written and I enjoyed reading it. I strongly support its publications – I feel the paper is essentially acceptable for publication already.

*Reviewer #3 (Recommendations for the authors):*

Figure 1d: I could not find at what stage of fermentation these measurements were done (probably day 14?).

Figure 1e: these results are not discussed in the text. The choice to separate bacteria and fungi does not seem intuitive to me because both organism classes will jointly determine the concentration of metabolites. Would not separating them in the analysis potentially reduce the percent variance explained?

Figure 1: Since authors only present endpoint measurements (the time of which was not very clear from the text) for the biochemical and microbiological composition of KT, what are their thoughts on the dynamics of these parameters? Could the differences between samples be explained just by differences in the speed of broth colonization/fermentation or what are the reasons to expect stabilized readout at the time of measurement?

The authors mention that similar ratios of B and Y CFUs in co-cultures represent the ratio in original KT. However, I could not find CFU data for total B and Y CFUs in the KT.

line 163: It is a very interesting result that B2 and B4 bacteria grew in combination with fungi but not on their own. The authors explain this in: 'bacteria varied greatly in sucrose utilization'. However, is it possible that yeast provides bacteria with other growth-limiting resources, not necessarily digesting sucrose to monosugars? On a more philosophical note, how does the choice between yeast-dependent and yeast-independent bacteria play into what is a representative microbial core?

Figure 3d: it is a little bit confusing to observe pellicle composition data for days 1-5 before any pellicle is formed. Perhaps adding a comment in the legend or revising the graph could be considered?

Do I understand correctly that ratios AND abundance of Y1 and B2 in pellicle remain relatively stable following its formation on day 6? I conclude this from supplementary figures 1d,e and methods indicating that pellicles were dissolved in equal volumes. If correct, this implies that pellicle dry weight increase is due to extracellular matrix formation and not growing microbial biomass. I feel that talking about ratios only leaves out this important finding about KT growth dynamics.

Figure 3k,l: What does mg/GAE and mg/QE stand for?

Line 354: For the 10 species experiments, I am left with the question of how many of them prevailed by the end of the fermentation? If one of the yeast (or bacteria) outcompetes the other four species, it may not be possible to appropriately conclude similarities between core and a more complex community.

Line 377: 'or crease' typo

---

## [Author Response]

Essential revisions:Your revised submission must address these two points resulting from the consultative review process. For specific guidance on both points, please see the individual reviews. Other points brought up by the reviewers do not have to be addressed in your point-by-point response.(1) Clearer, more informative figures and better explanations of figures.

We thank the reviewers for taking time to read our manuscript and provide valuable

comments. Following the suggestion, we have examined all figures and captions, and revised some of them to provide better illustrations and explanations. For Figure 1D, we specified in the caption the time (day 14) at which measurements were conducted. For Figure 1E, we provided an explanation of separate analysis for bacteria and fungi in the main text. For Figure 2B, we supplemented the source data for the absolute concentrations of the metabolites. For Figure 3A-B, we replaced growth rates with the absolute species abundances in semilog plots. For Figure 3D, we used dashed lines to indicate microbial population ratios before observing the pellicle formation. For Figure 3K, we introduced detailed definitions about the units mg/GAE and mg/QE in the corresponding caption. For Figure 4, we changed the colors of the lines to provide better contrasts. Additionally, we added a schematic diagram (Figure 7) to summarize the overall concept and procedure of our complexity-reduction approach for microbiome dissection. Together, we believe that these revisions greatly improve the clarity of the figures.

(2) Discuss limitations of the approach (e.g. in vitro fermentation communities are easier as the host is not a factor; the original community is already quite simple compared to other types of communities).

This is a highly constructive comment. We have now included a discussion about the

limitations and future efforts of our approach in the Discussion section as follows:

“In this study, we successfully identified and utilized a two-species consortium as the core for systematic analysis of the KT microbiome. However, such a strategy may become less straightforward for many native microbiomes, such as those in the soil and in the human gut, due to their remarkable degree of complexity and their close association with the corresponding hosts (e.g., plant and human). Additionally, the KT microbiome studied here was driven primarily by pairwise interactions, another factor that made it possible to use the two-species ecosystem as an approximate of the entire community. For those involving higher-order interactions, microbial social interactions are much more complicated, and it needs significantly more efforts to screen and determine valid cores. In these cases, whether and to what extent a relatively simple core can be defined will depend on the specific starting microbiome, as well as the target functions of interest. In addition, a single microbiome in principle can possess multiple cores depending on different selection criteria, such as abundance, temporal pattern and function. Future efforts in these aspects are needed to fully realize the power of this community analysis strategy.”